

# Soccer games and record-breaking PM$_{2.5}$ pollution events in Santiago, Chile.

Rémy Lapere[1], Laurent Menut[1], Sylvain Mailler[1], and Nicolás Huneeus[2]

[1]Laboratoire de Météorologie Dynamique, Ecole Polytechnique, Palaiseau, FR
[2]Departamento de Geofísica, Facultad de Ciencias Físicas y Matemáticas, Universidad de Chile, Santiago, Región Metropolitana, CL

**Correspondence:** Rémy Lapere (remy.lapere@lmd.polytechnique.fr)

**Abstract.** In wintertime, high background concentrations of atmospheric fine particulate matter (PM$_{2.5}$) are commonly observed in the metropolitan area of Santiago, Chile. Hourly peaks can be very strong, up to ten times average levels, but have barely been studied so far. Based on atmospheric composition measurements and chemistry-transport modeling (WRF-CHIMERE), the chemical signature of sporadic skyrocketing wintertime PM$_{2.5}$ peaks is analyzed. This signature and the

timing of such extreme events traces their origin back to massive barbecue cooking by Santiago's inhabitants during international soccer games. The peaks end up evacuated outside Santiago after a few hours but trigger emergency plans for the next day. Decontamination plans in Santiago focus on decreasing traffic, industrial and residential heating emissions. Thanks to the air quality network of Santiago, this study shows that cultural habits such as barbecue cooking also need to be taken into account. For short-term forecast and emergency management, cultural events such as soccer games seem a good proxy

to prognose possible PM$_{2.5}$ peak events. Not only this result can have an informative value for the Chilean authorities, but a similar methodology could also be reproduced for other cases throughout the world in order to estimate the burden on air quality of cultural habits. In particular, the present study shows that investigating the atmospheric composition in large cities during major events is key for the design of effective air pollution mitigation policies.

## 1 Introduction

Santiago, the capital city of Chile (33.5°S, 70.5°W, 570m a.s.l.) regularly faces high levels of fine particulate matter (PM$_{2.5}$) pollution in winter. The city is located in a confined geographical basin surrounded by the Andes cordillera in the East, a coastal range in the West, and transversal mountain chains in the South and North (Rutllant and Garreaud, 1995). The induced poor ventilation in wintertime combined with significant anthropogenic emissions lead to high background levels of PM$_{2.5}$

(Barraza et al., 2017; Mazzeo et al., 2018) as well as peak events (Toro et al., 2018). Hourly surface concentrations can reach up to 600$\mu$g/m$^3$ in the Western part of the city according to the local air quality monitoring network. Between June and July 2016 records show that the station of Pudahuel saw only seven days with an average PM$_{2.5}$ concentration below the 25$\mu$g/m$^3$





24-hour mean standard defined by the World Health Organization (World Health Organization, 2006). 7 million people live in the metropolitan area and are exposed to such atmospheric pollution. The associated life expectancy reduction caused by PM$_{2.5}$ inhalation ranks Chile among the countries with air pollution issues (Energy Policy Institute at the University of Chicago, 2017). In this respect, atmospheric decontamination plans were designed by local authorities in the recent years (Gallardo et al., 2018; Ministerio del Medio Ambiente, 2012). However the source and impacts of extreme peak events as well as the benefits of their mitigation are relatively unknown.

Several studies have been conducted to improve the PM$_{2.5}$ concentration forecast system in Santiago (Rutllant and Garreaud, 1995; Saide et al., 2016; Mazzeo et al., 2018). However, none of them describe the sharp sporadic peaks observed some years in June and July nor explain their origin, while their impact is substantial. Policies designed to mitigate chronic air pollution are expected to have large benefits both in terms of public health and economy on the long run (Salinas and Vega, 1995; Bell et al., 2005; Valdés et al., 2012). However, acute health effects of strong, time-limited PM$_{2.5}$ events are also known to be significant in Santiago, with increases in respiratory emergency and pneumonia visits within 2 days after such a peak (Ilabaca et al., 2011). Government reports also provide evidence that highly polluted conditions affect the local economy, and estimate the net benefit of compliance with PM standards to several millions of US dollars (Ministerio del Medio Ambiente, 2012). This study combines the automated air quality monitoring network of Santiago and chemistry-transport modeling to describe and identify the source of recent short-lived PM$_{2.5}$ extreme events occurring in wintertime. The dispersion pattern of such an event in June 2016 is also modeled.

Section 2 presents the data and model configuration used in this study. Section 3 describes the outcomes of both the data analysis and the chemistry-transport simulations regarding the identification of the origin of the extreme events considered. Section 4 discusses the hypotheses underlying the conclusions, which are detailed in Section 5.

## 2 Data and methods

### 2.1 Observation data

Time series of hourly surface measurements of meteorology and air quality are extracted from the automated air quality monitoring network of Santiago (SINCA - https://sinca.mma.gob.cl/index.php/region/index/id/M). The distribution of the monitoring stations can be seen in Fig. 4 for instance. This network uses beta ray attenuation technology (Met One Instruments' Model BAM-1020) for PM$_{2.5}$ concentrations measurements, gas-phase chemiluminescence (Thermo Scientific Model 42i) for NO$_{X}$, and infrared photometry by gas filter correlation (Thermo Scientific Model 48i) for CO. Vertical meteorological profiles used for the validation of the simulations were provided by the Dirección Meteorológica de Chile. Ceilometer back-scattering profiles were measured and provided by the University of Chile.



## 2.2 Model set-up

The chemistry-transport simulations are based on the combination of WRF mesoscale weather numerical model, HTAP anthropogenic emissions inventory and CHIMERE chemistry-transport model. The simulation domains are described in Fig. 1. The
meteorological conditions are simulated using the Weather Research and Forecasting model from the US National Center for Atmospheric Research (Skamarock et al., 2008). The model configuration used in this study to simulate and reproduce observed meteorological conditions are presented in Table 1. The model was applied to 46 vertical levels up to the highest elevation of 50hPa, in a two-way nested fashion, with 1-2-1 smoothing. Initial and boundary conditions used are FNL analysis from the Global Forecast System (NCEP, 2000). Land-use and orography are based on the modified IGBP MODIS 20-category database
with 30sec resolution (University of Maryland, 2010). The simulated period is June 1$^{st}$ to July 15$^{th}$ 2016, with a spin-up period from June 1$^{st}$ to June 15$^{th}$. CHIMERE is an Eulerian 3-dimensional regional Chemistry-Transport Model, able to reproduce gas-phase chemistry, aerosols formation, transport and deposition. In this study the 2017 off-line version of CHIMERE is used (Mailler et al., 2017). The configuration used for this study is described in Table 1. Land-use and orography data is the same as for WRF. As background anthropogenic emissions, the HTAP V2 dataset is used which consists of 0.1° gridded maps of air
pollutant emissions for the year 2010 (Janssens-Maenhout et al., 2015). A downscaling is applied to this inventory based on land-use and demographics characteristics, and monthly emissions are split in time down to daily/hourly rates following the methodology of (Menut et al., 2013).

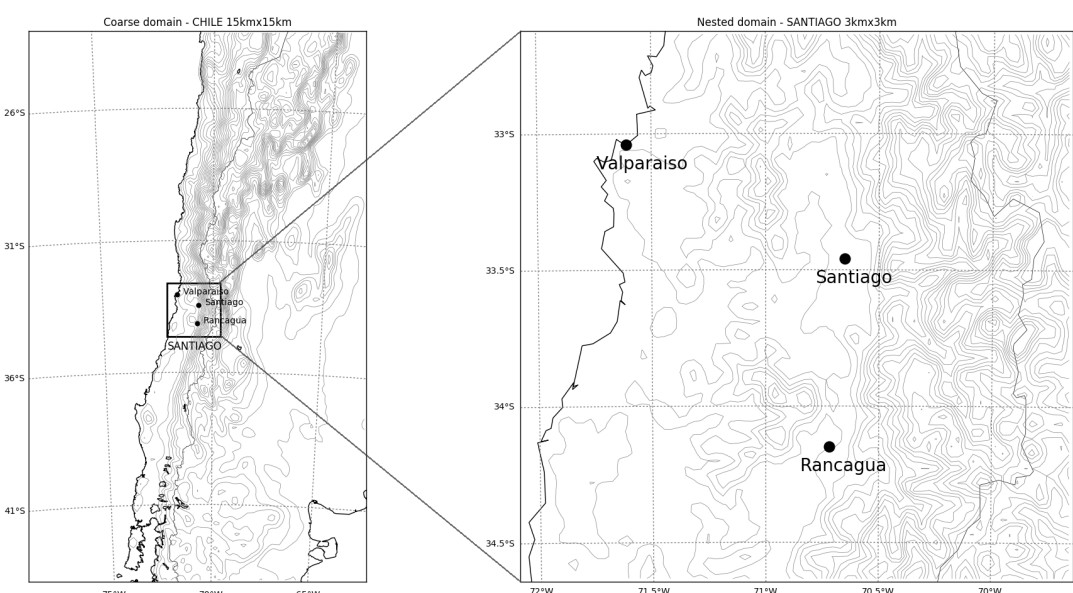

**Figure 1.** Left: coarse simulation domain at 15km resolution. Right: nested domain at 3km resolution. 250m contour levels shown are interpolated from the modified IGBP MODIS 20-category database with 30sec resolution (University of Maryland, 2010).





## 2.3 Simulation validation

Simulation scores are gathered in Tables 2, 3 and 4. Two stations downtown are used to validate the simulated near-surface meteorology - Table 2. Biases (MB) for temperature are around +/-1°C with correlations (R) around 0.85. The model is a little too dry with relative humidity biases between -12% and -16% but mostly reproduces the diurnal cycle with correlations of 0.62 and 0.7. 10m wind speed time series is fairly well reproduced with biases of -0.08m/s and 0.23m/s respectively and correlations of 0.56 and 0.7. Vertical meteorological profiles scores are shown in Table 3. For the three days presented the

statistics are satisfactory. Background concentrations of $PM_{2.5}$ are quite well reproduced by the model. Table 4 gathers the scores for $PM_{2.5}$ for some stations, between June $28^{th}$ and July $15^{th}$ so as to avoid the peaks that do not represent business as usual conditions. Mean biases are less than $5\mu g/m^3$ and correlations are between 0.45 and 0.63, which are decent values. The representation of wind by the model for synoptic meteorological stations in Santiago, is described in Table 3 and Fig. 2. Generally speaking the model behaves well for 10m values and profiles of wind speed and direction, so that transport should

be realistically represented.

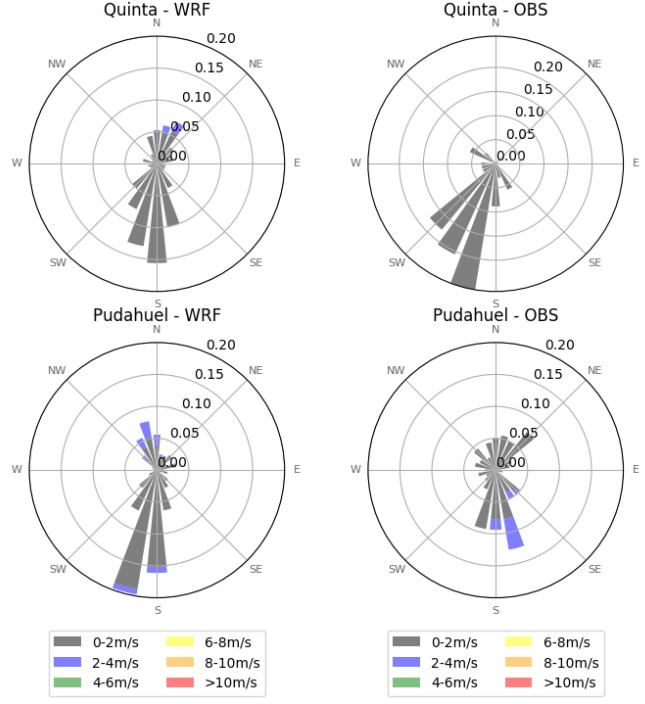

**Figure 2.** Modeled (left) and observed (right) wind rose between June $15^{th}$ and $30^{th}$ 2016 - synoptic stations Pudahuel and Quinta Normal

The modeling setup is based on WRF-CHIMERE, with a horizontal resolution of 3km, and emissions are downscaled from a dataset originally at a 0.1° resolution. At the scale of a city such as Santiago, which is roughly 20km by 20km, such a resolution might seem too coarse to capture the observed heterogeneity. However, the previous analysis shows that the meteorological conditions are well reproduced by the model, and the spatial distribution of $PM_{2.5}$ concentrations also accounts

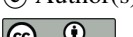



for the observed heterogeneity (see Fig. 10 for instance). (Mazzeo et al., 2018) used a similar setup with a 2km resolution for a sensitivity analysis to traffic and residential heating emissions in Santiago, yielding similar performances. Comparable CHIMERE simulations are performed for purposes of air quality operational forecast in France, which performance at small scale is acknowledged in the literature, provided emissions have appropriate magnitudes (Petit et al., 2017; Shaiganfar et al., 2017).

# 3   Results

## 3.1   PM$_{2.5}$ peaks description

The following analysis is based on the data provided by the "Sistema de Información Nacional de Calidad del Aire" (SINCA) network of surface air quality sensors distributed in the metropolitan area of Santiago (Ministerio del Medio Ambiente, 2018). The time series of PM$_{2.5}$ concentrations for each of the 11 stations of this network for June and July 2016 are illustrated
in Fig. 3. Two skyrocketing peaks, up to ten times higher than the average background concentrations, occurred at several stations during the nights between June 18[th]/19[th] and 26[th]/27[th]. These two peaks reached all-time record breaking levels for some stations in the city according to the available SINCA time series. Other stations show less extreme peaks, which is representative of the rich dynamics of particulate matter within Santiago (Toledo et al., 2018). Understanding the origin and modeling the dispersion of these time-limited, very sharp events is the purpose of this study. For the following analysis, the
peak on June 26[th]/27[th] is considered, although the analysis and results are the same for June 18[th]/19[th]. Its spatial evolution can be found in Fig. 4, showing that the episode starts simultaneously at several air quality stations at around 8pm, levels decreasing back to regular values the next morning. The Western part of the city seems much more affected by the event than the Eastern part. This comes from the diurnal wind cycle in Santiago, that features prevailing easterlies during night time contributing to renew air masses in this part of the city (Rutllant and Garreaud, 1995).

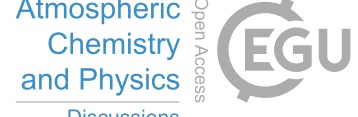

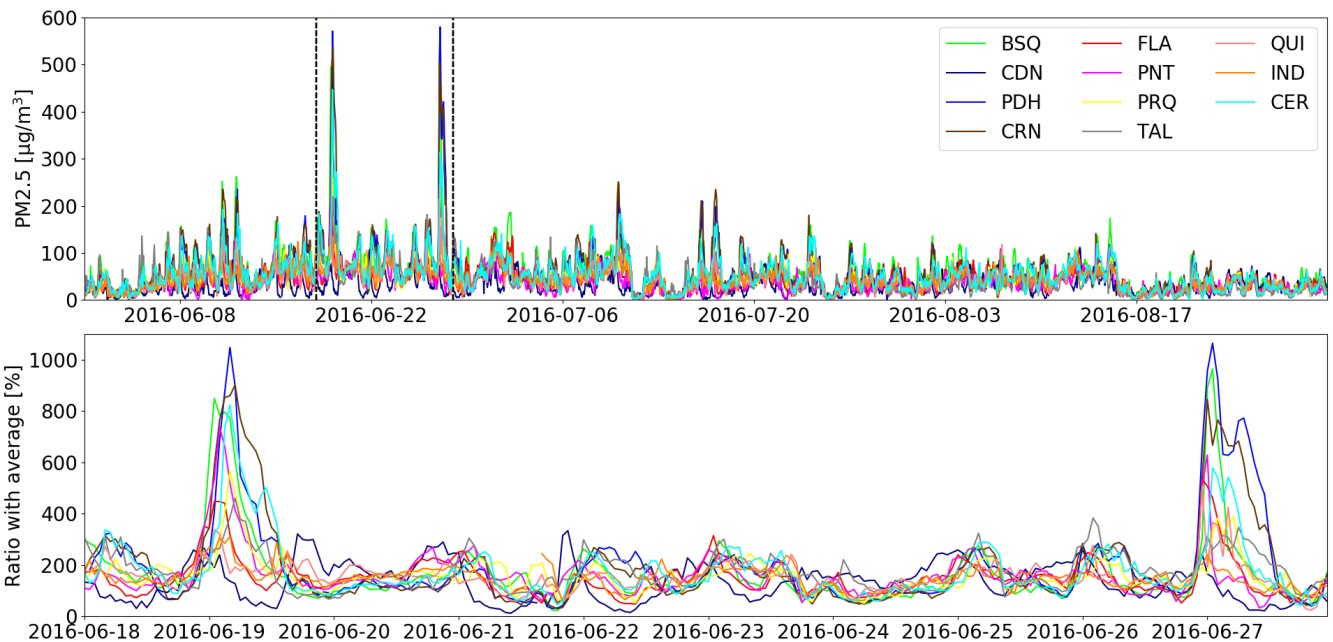

**Figure 3.** Top: time series of hourly PM$_{2.5}$ concentration between June 1$^{st}$ and August 31$^{st}$ 2016 for the 11 stations of the air quality network of Santiago. Bottom: ratio between hourly PM$_{2.5}$ and average over the summer, zoomed between June 18$^{th}$ and June 28$^{th}$ (dashed vertical lines in top graph).





(a) 06/26 - 18h

(b) 06/26 - 20h

(c) 06/26 - 22h

(d) 06/27 - 00h

(d) 06/27 - 07h

(e) 06/27 - 09h

**Figure 4.** Observed PM$_{2.5}$ hourly concentrations during the peak on June 26$^{th}$ compared to its average over summer 2016 - blue <= 200%, green <= 300%, yellow <= 400%, orange <= 500%, red >500%. Map background layer: World Shaded Relief, ESRI 2009.





The meteorological conditions observed at the location of the strongest peak during our period of interest are shown in Fig. 5. Red lines correspond to the measurements between June 26th 6 a.m. and June 27th 5 a.m. (local time) which comprises a peak event. Figures 5a and 5b show that at the location of the strongest PM$_{2.5}$ peak, the surface temperature and relative humidity cycles over the duration of the event are very close to the average over the month (black line). Wind speed - Fig. 5c - is a little slower than average on the whole but remains in the range of the 1st and 3rd quartiles of ventilation conditions (dashed white

lines). Ceilometer profiles and boundary layer height (BLH) estimation based on the methods from (Muñoz and Undurraga, 2010) derived for each day at 3 p.m. - white diamonds in Fig. 5d - indicate that the mixed layer is rather shallow around the peak but not shallower than on the 22nd/23rd for instance, which did not feature such an event. On June 26th, the mixed layer is 430m high, which is close to the average height over the period, and much higher than the minimum value (260m) obtained on June 21st.

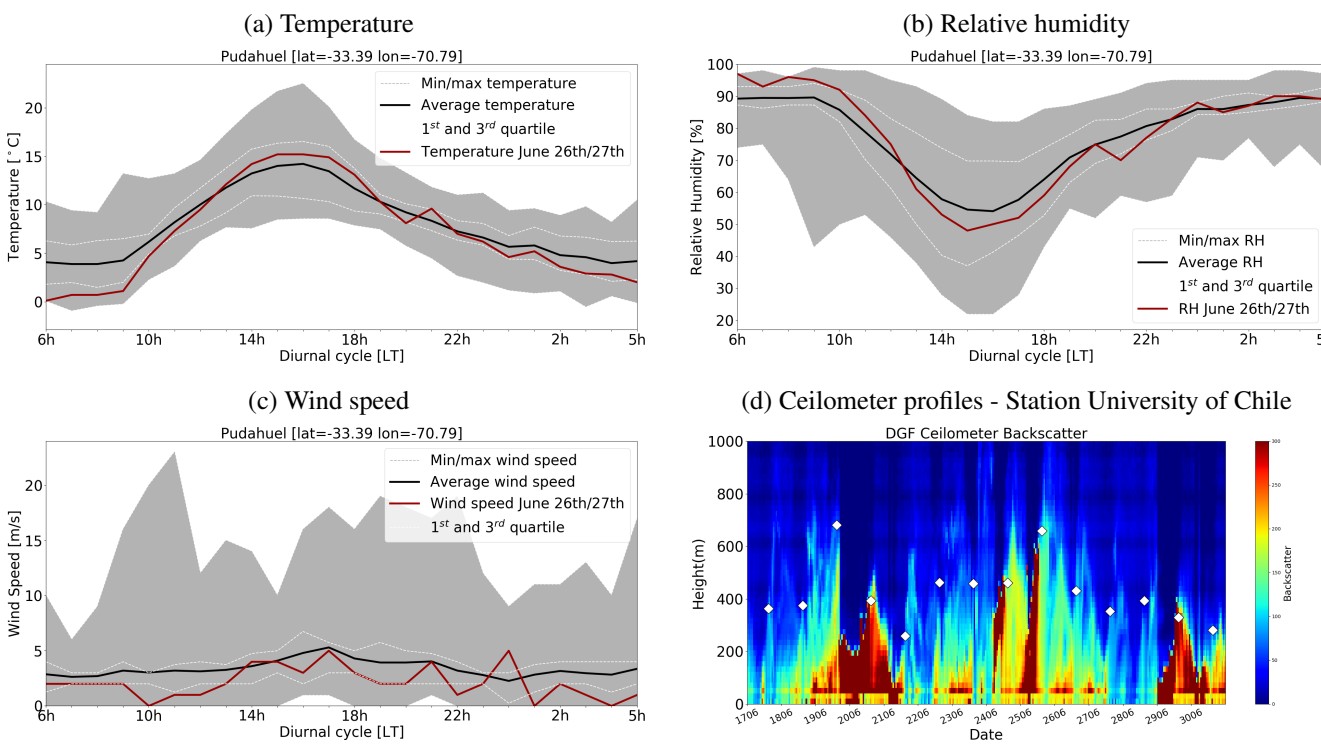

**Figure 5.** Meteorological conditions around June 26th/27th PM$_{2.5}$ episode for the synoptic station Pudahuel. (a),(b),(c) average (black line), minima and maxima (gray area), 25% and 75% quantiles (white dashed line) between June 15th and July 15th 2016. The red lines show the values for June 26th 6 a.m. to June 27th 5 a.m. (d) hourly backscatter profiles from June 17th through June 30th 2016 and mixed layer height at 3 p.m. (white diamond markers).





Hence, measurements show that the meteorological conditions during the peak are not very different from other days of the period, when no PM peak was recorded. Thus, meteorological conditions are not to consider as the main forcing for the PM$_{2.5}$ event although they are favorable for it to appear, which is usual for the season in the area of Santiago.

Besides meteorology, advection of a PM plume over the city could be a candidate cause for such events. For instance, wildfires occurring in the forests surrounding the city occasionally explain major peaks of particulate matter in Santiago
(Rubio et al., 2015; de la Barrera et al., 2018), but these events mostly take place in austral summer, and no such event was reported during our period of study. In addition, the strong concentration gradients observed between nearby stations (Fig. 3 and 4) makes advection unlikely responsible for these events.

Once meteorology and transport are ruled out as root causes, high local emissions must then be underlying the very strong concentrations recorded.

**3.2    Chemical signature and source identification**

In order to identify the type of source involved in the sporadic peaks, Fig. 7 shows a scatter plot of PM$_{2.5}$, NO$_X$ and CO hourly surface concentrations from June 15$^{th}$ to June 30$^{th}$ 2016, for Pudahuel air quality station. Red dots correspond to PM$_{2.5}$ concentrations higher than $200 \mu g/m^3$, blue dots to concentrations below that value. Out of simplicity, from now on we will refer to the former as PPE (PM peak events) and to the latter as PRS (PM regular situation). Two different regimes can be identified:
for PPE, the NO$_X$/CO ratio is around 4.6%, while for PRS, it is approximately 14%. The same goes for the NO$_X$/PM$_{2.5}$ ratio, with values around 73% for PPE, and 502% otherwise. Given the short-lived character of the events considered, such different concentration ratios can be related to different emission factors as discussed in Sect. 4. Thus, two different types of source are involved in the two situations (PPE/PRS) considered.

(Mazzeo et al., 2018) found a NO$_X$/PM$_{2.5}$ ratio of 526% for Pudahuel station in July 2015, without peak events. Using the
same methodology and combining data from July 2015 and 2016, without peak events, we recover a concentration ratio around 557% for this same station. For NO$_X$/CO we find it is around 14% (see Fig. 6). These two values correspond to the background pollution situation in this part of the city. They are comparable to what can be observed in PRS in Fig. 7 with an average ratio of 502% for NO$_X$/PM$_{2.5}$ and 14% for NO$_X$/CO. Therefore, the PM regular situation fairly corresponds to the background situation in Santiago in winter. Peak events (PPE - red dots) however show very different ratios that do not coincide with the background
situation, hence pointing to another specific source. Major contributors to atmospheric pollution in Santiago are traffic (39%), industry (18%) and residential heating (20%) (Barraza et al., 2017). Based on the current EURO 5 legislation for car engines emissions in place in Chile (Ministerio del Medio Ambiente, 2017) and the vehicles fleet in Santiago (Instituto Nacional de Estadísticas, 2016), the expected emission ratio from traffic yields around 12% for NO$_X$/CO and 1680% for NO$_X$/PM$_{2.5}$. This does not match with the PPE signal, and traffic being the driver of the background pollution, the concentration regime would
not be that different. Residential heating by domestic combustion of wood and/or fossil fuel is expected to have slow variations in time, depending essentially on outside temperature (Saide et al., 2016), which as discussed above was not particularly colder on peak days than on other days, permitting to rule out residential heating as the main contribution to these short-lived peaks.





Industrial emissions are also generally constant through time, excluding the possibility that they could be the main factor either. Other sources than usual must then cause these peaks.

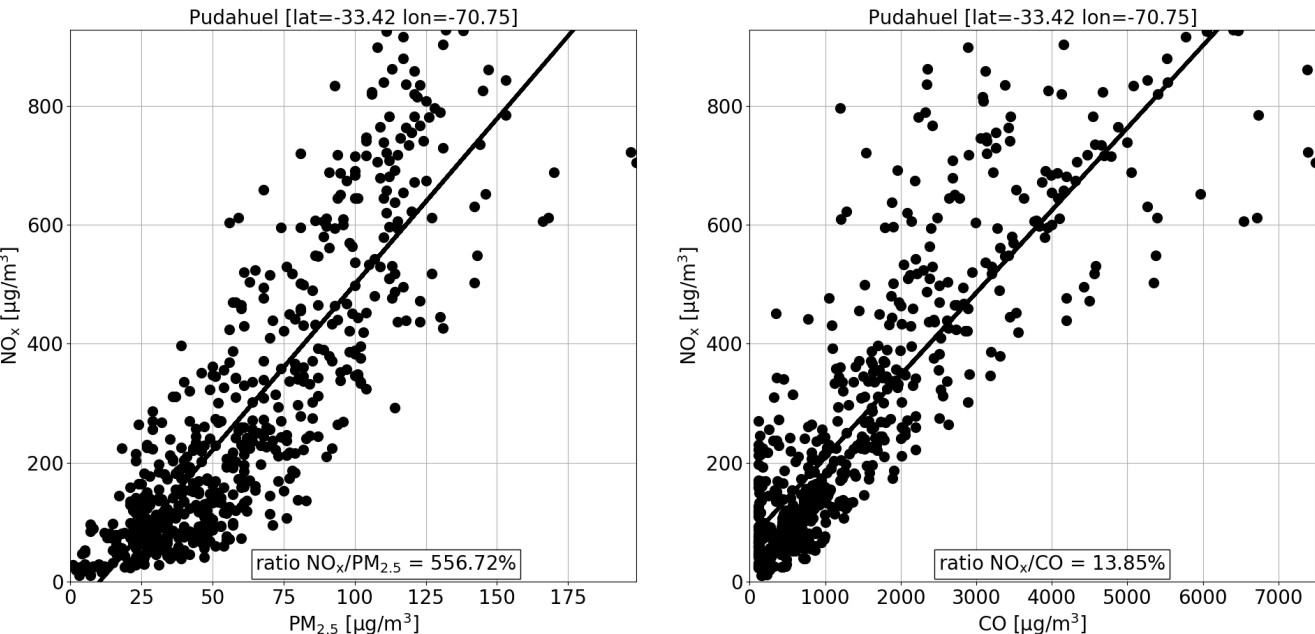

**Figure 6.** Background pollutants ratios for July 2015 and 2016 - Pudahuel station

Since the peak events considered occurred exclusively during evenings/nights, cooking emissions such as barbecues, which are a cultural habit in Chile, could be a candidate. Different studies estimate emission factors from barbecue cooking (charcoal only and including meat emissions), from which ratios of 1.4% (Vicente et al., 2018) to 2.4% (Lee, 1999) for $NO_X/CO$ and 40.5% (Vicente et al., 2018) to 61.5% (Lee, 1999) for $NO_X/PM_{2.5}$ can be derived. These numbers are not far from the 4.6% and 73% observed during PPE (Fig. 7), suggesting that, although the atmospheric composition is likely the result of multiple

sources, the main signal in the observed $PM_{2.5}$ concentrations corresponds to emissions from barbecues. As a first order approximation, we use emission ratios and concentration ratios equivalently. This assumption is examined in Sect. 4. However, one question remains: why would there be peaks of barbecue cooking in the nights of June 18[th] and June 26[th] specifically, rather than other nights during the studied period? Even more so given that barbecues are mainly a spring-summer activity in Chile.

Usually, barbecues (or "asados") are cooked when celebrating particular events in Chile. So as to gain statistical significance by including more events, the time period studied is expanded to winter 2015 and 2014 as well. As it turns out, during the month of June of these three years, 8 episodes with hourly $PM_{2.5}$ higher than $200\mu g/m^3$ were recorded at Pudahuel station, 5 of which peaked during international soccer games involving the Chilean national team. The three others were declared within 24 hours after a game. It is even clearer for peaks higher than $400\mu g/m^3$, all of them occurring at the exact same hour as the





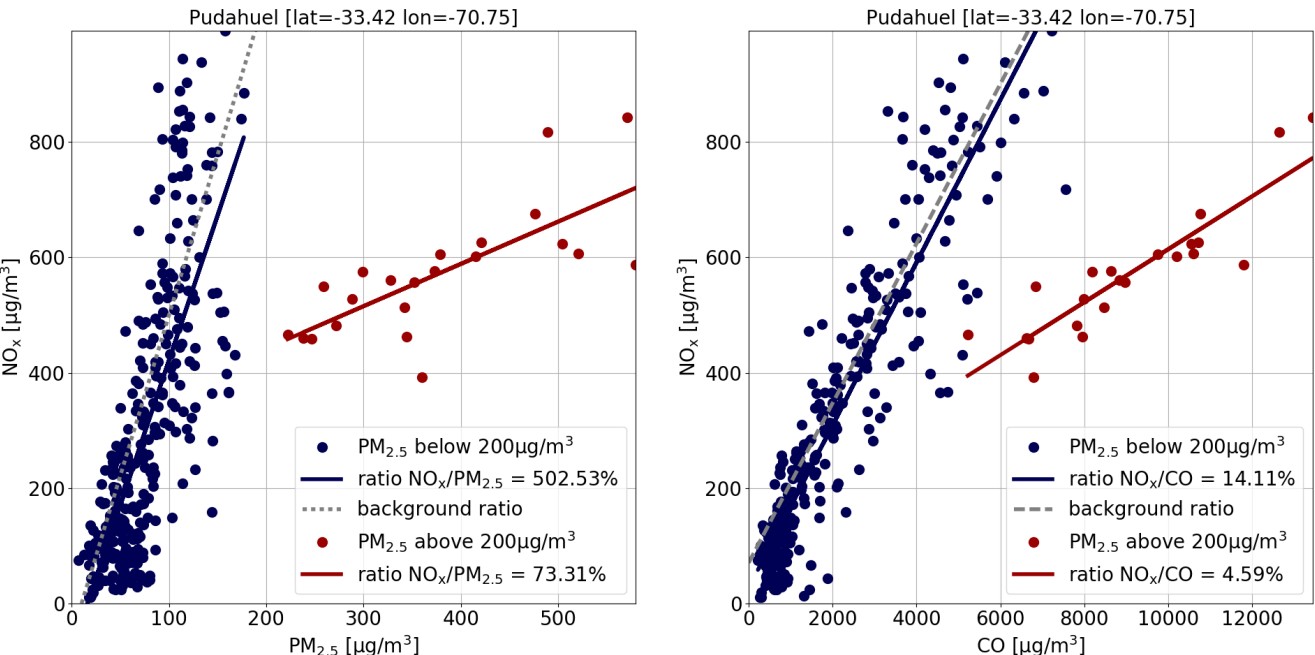

**Figure 7.** Observed $NO_X/CO$ and $NO_X/PM_{2.5}$ concentration ratios at Pudahuel station (PDH). Blue dots are for $PM_{2.5}$ concentrations below $200\mu g/m^3$ - red dots are for $PM_{2.5}$ concentrations above $200\mu g/m^3$ - gray line represents the background ratio in July 2015/2016 - blue and red lines correpond to linear regressions for each data set.

kick-off of a soccer game when the next day is not a working day - see Fig. 8: red diamonds represent Chile games kick-off during the 2014 world cup, 2015 Copa América and 2016 Copa América. In addition, for the recent years without major soccer championship involving the team of Chile (i.e. 2010, 2012, 2013, 2017), the maximum hourly $PM_{2.5}$ concentration at Pudahuel over the same period of the year was $260\mu g/m^3$. In 2011 another Copa América was played and values reached up to $360\mu g/m^3$.

This correlation is likely not coincidental. Over the months of June 2014, 2015 and 2016, the observations show 12 days recording "pre-emergency" conditions (24-hour $PM_{2.5}$ concentration above $110\mu g/m^3$). Among these 12 days, 10 dates coincide with a soccer game of the national team or the day after such a game (games being played at nights, the peak affects both the game day and the day after). The 3 periods total 90 days, 15 of which were game days. Based on combinatorics, the probability that soccer games and $PM_{2.5}$ pre-emergency levels coincidentally occur with a proportion of at least 10 for 12 can be expressed as in Eq. (1). This corresponds to randomly drawing 15 days out of the 90 available, and obtaining at least 10 peaks. As a result, the probability that the correlation between $PM_{2.5}$ peaks and soccer games is purely coincidental is 0.002%. Thus we can be confident that there is actually a significant correlation between these two types of events, through massive barbecue cooking during games.

$$\mathbb{P}_{\mathrm{rand}} = \frac{\binom{12}{10}\binom{78}{5} + \binom{12}{11}\binom{78}{4} + \binom{12}{12}\binom{78}{3}}{\binom{90}{15}} = 2.10^{-5} \tag{1}$$



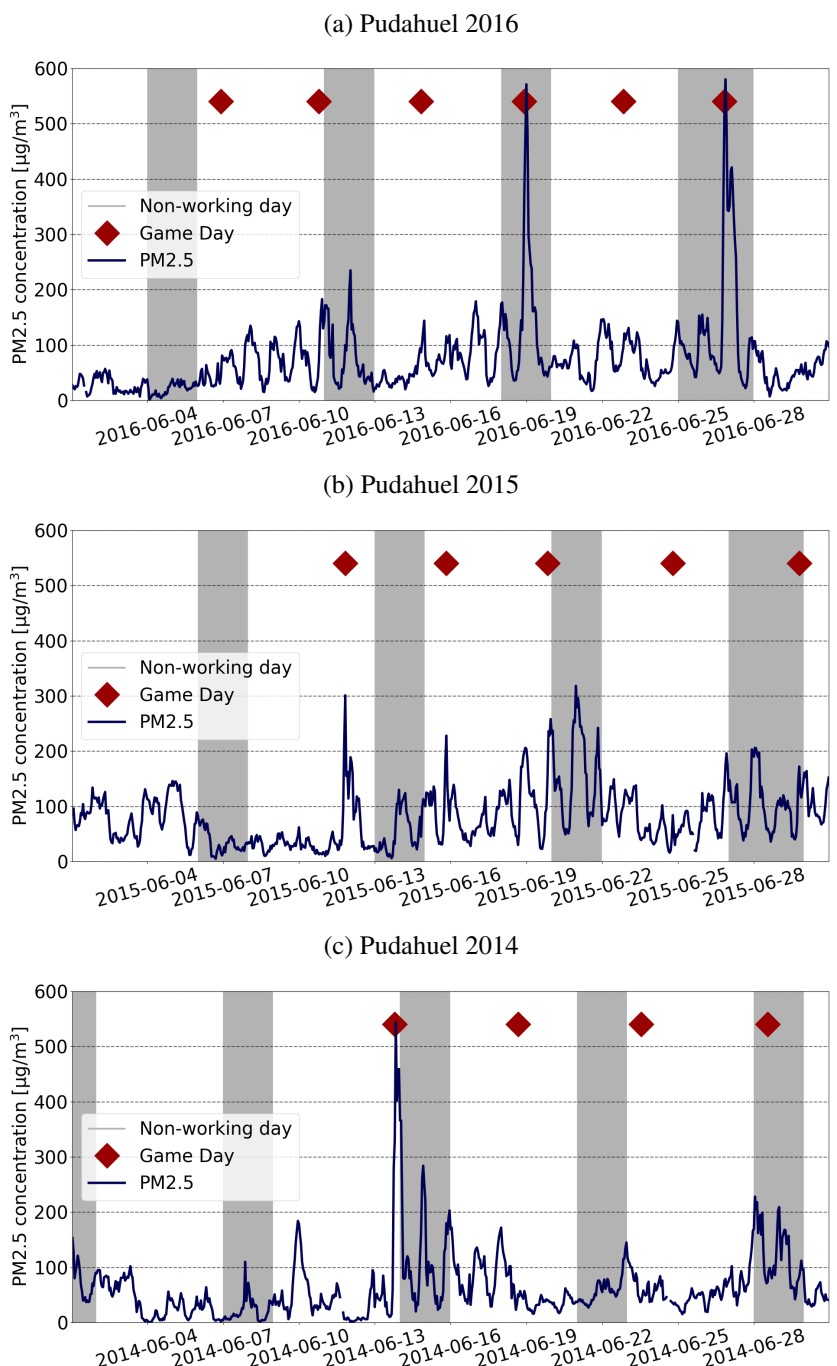

**Figure 8.** PM$_{2.5}$ peak events coincidence with soccer games. (a) Hourly PM$_{2.5}$ concentrations at Pudahuel monitoring station June 2016 (b) Hourly PM$_{2.5}$ concentrations at Pudahuel monitoring station June 2015 (c) Hourly PM$_{2.5}$ concentrations at Pudahuel monitoring station June 2014.





The observations made in Fig. 7 are also valid for other stations throughout the city. Figure 9 shows the $NO_X/CO$ and

$NO_X/PM_{2.5}$ ratios for three other locations in Santiago, distinguishing between the hours around the games of June 18[th] and

June 26[th] (red dots), and the other data for the month (blue dots). Then again, two different regimes are observed, the one

during games being attributable to barbecues based on the same analysis as previously done, although barbecues seem to be

even more dominant in Pudahuel. In summary, we have shown here that major peak events of $PM_{2.5}$ in Santiago are correlated

with soccer games played on the evening before a non-working day, and such conditions can be tied to massive barbecue

cooking throughout the city, given the chemical footprint observed at that times.

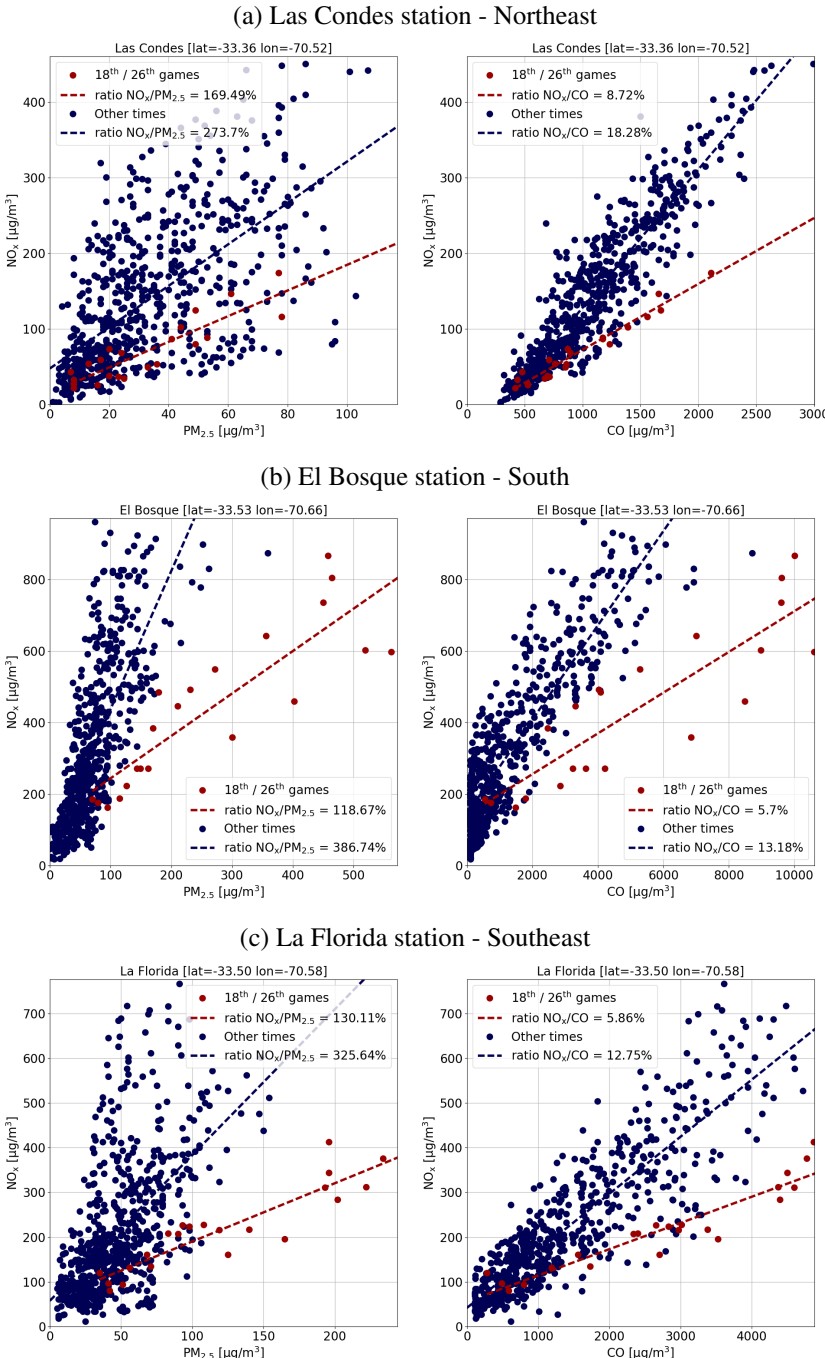

**Figure 9.** Observed NO$_X$/CO and NO$_X$/PM$_{2.5}$ concentration ratios at 3 stations in June 2016. Blue dots are observations during the June 18$^{th}$ and June 26$^{th}$ soccer games. Red dots correspond to the rest of the data. Blue and red dashed lines correpond to linear regressions for each case.





### 3.3 Transport

These barbecue peaks, although generating large amounts of $PM_{2.5}$, last only a few hours. The termination of one of these events is studied hereafter. A chemistry-transport simulation is run with WRF (Skamarock et al., 2008) and CHIMERE (Mailler et al., 2017) for the austral winter 2016 (see 2 for the details). A first baseline simulation aiming to reproduce observed background concentrations of pollutants is performed using HTAP anthropogenic emissions inventory (Janssens-Maenhout et al., 2015). This inventory does not account for sporadic emissions such as the ones studied here. Despite a good performance in reproducing the observed meteorology and background atmospheric composition (see 2 section, Tables 2, 3, 4), the model does not produce peak events on June 18th and June 26th (dashed black line in Fig. 10). This reinforces the idea that strong sporadic emissions are actually at play rather than extreme weather conditions. In a second simulation, all other things remaining equal, strong additional sources of $PM_{2.5}$ are added all over the city based on population density, and plugged into CHIMERE in order to account for barbecues being cooked on June 26th.

A survey conducted before the final game of the 2016 Copa América estimated that 29% of Santiago's inhabitants would cook a barbecue during the game (Panel Ciudadano de la Universidad del Desarrollo, June $26^{th}$ 2016). As a first order approximation, considering only the adult population and assuming that this is a group activity gathering on average 7 people, we estimate that this corresponds to 100,000 fires that were lit at the time of the game. Based on PM emission rates estimated by the US Environmental Protection Agency (Lee, 1999) at around 20g/hr on average, with variations depending on the type of meat cooked, the expected additional emission of $PM_{2.5}$ would be a total 2 tons per hour for the whole region. In a heavily populated area such as El Bosque, this represents an additional signal 15 times higher than the $PM_{2.5}$ emission rate used in the baseline simulation. We acknowledge the multiple sources of uncertainty in this estimation. However our goal is to explore barbecues as a potential significant source of $PM_{2.5}$ pollution, which only requires orders of magnitude given the strength of the signal. Barbecues are assumed to last 3 hours starting 1 hour before the game kick-off. The estimated additional emissions are plugged in CHIMERE for the peak simulation (gray line in Fig. 10).

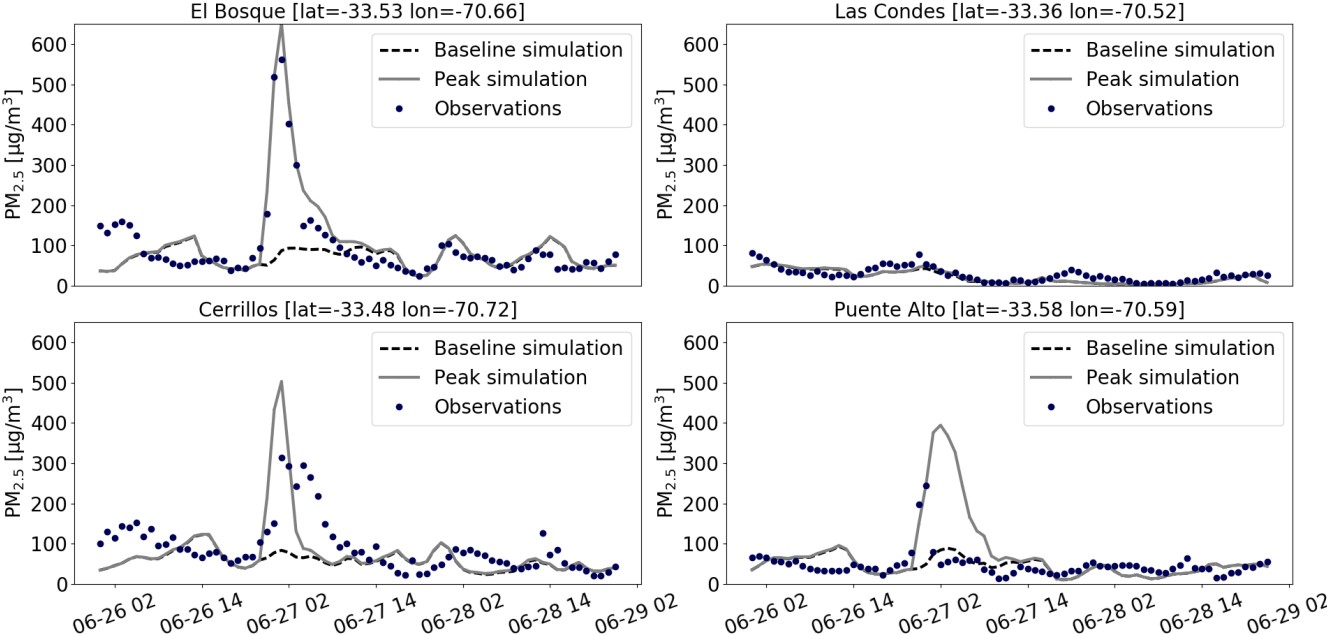

**Figure 10.** Observed (blue dots) and simulated (baseline simulation in black dashed line, peak simulation in gray) PM$_{2.5}$ surface concentrations time series at 4 stations in Santiago, around the peak episode of June 26$^{th}$ 2016

The resulting PM$_{2.5}$ concentration time series in Fig. 10 shows that the observations are well reproduced using this proxy in the South (El Bosque) and Northeast (Las Condes). Although the magnitude is a little overestimated but not far off, the time evolution of concentrations is less well reproduced in the West (Cerrillos) where the peak is too short, and in the Southeast (Puente Alto) where it lasts too long. This is possibly due to the representation of slope winds by the model: too weak easterlies coming from the Andes in the simulation would result in not enough ventilation in the East (i.e. a peak too long) and not enough accumulation in the West (i.e. a too short-lived event). Not many observations are available to investigate this hypothesis. Generally speaking, the peak simulation comforts the magnitude of our estimate of 100,000 barbecues cooked during the game, or an additional 6 tons of PM$_{2.5}$ emitted in total for the area.

Based on this simulation, the transport of the particles generated by the barbecue events can also be studied. Figure 11 shows the difference in PM$_{2.5}$ concentration between the two simulations aforementioned. Concentrations at 60m above ground level are considered in order to get rid of the signal of emissions. The simulation results in an evacuation of the particles towards the Southwest of Santiago, a few hours after the onset of the event. Then again the episode is short-lived in the city, but has impacts on adjacent areas several hours later and several kilometers away from the emission site. In addition to the impact within the city, collateral effects outside Santiago must be studied in greater detail, such as a potential deposition of the particles on crops, with effects on yields, or adjacent glaciers, with radiative effects.



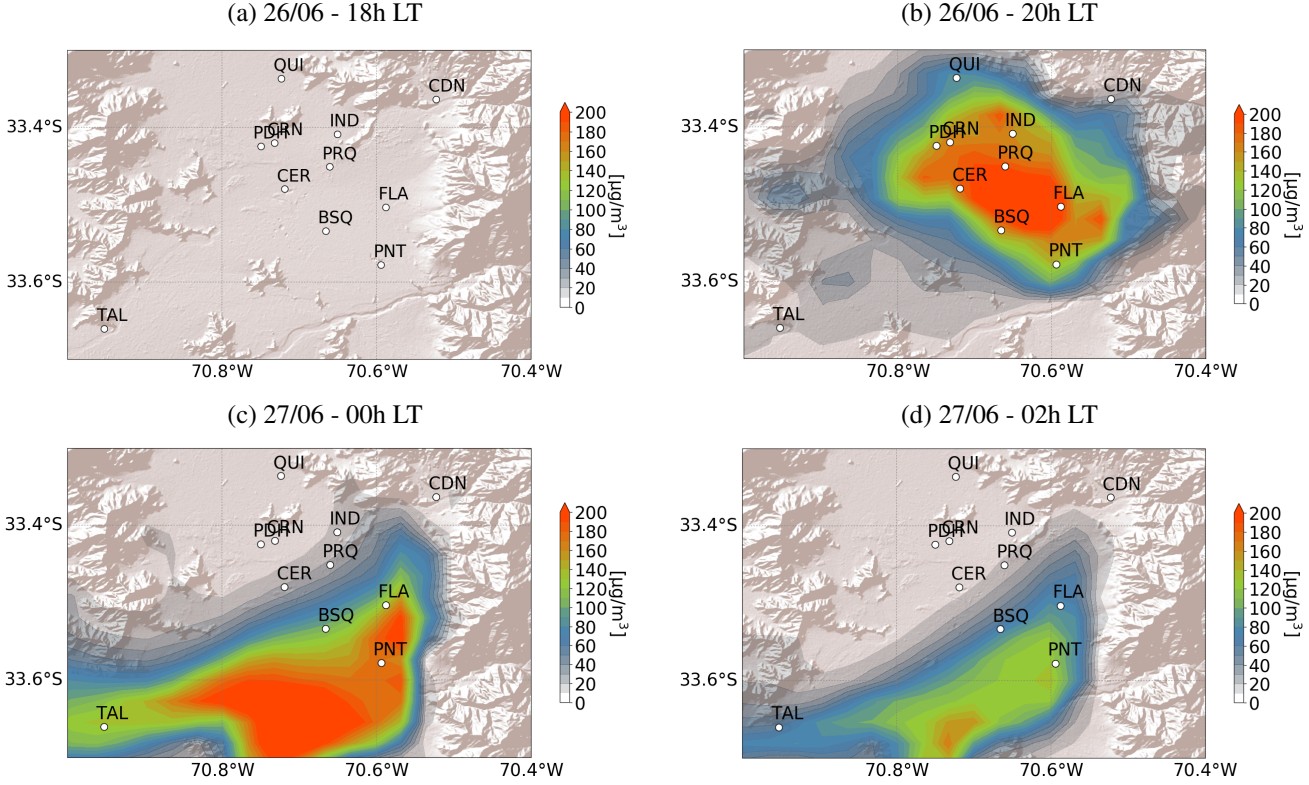

**Figure 11.** PM$_{2.5}$ concentration difference at 60m above surface between the peak event and baseline simulation in June 26$^{th}$/27$^{th}$. Positive values indicate excess concentrations in the peak event scenario compared to baseline. Map background layer: World Shaded Relief, ESRI 2009.

## 4 Discussion

In Sect. 3.2, the conclusions are based on the approximation that the concentration ratios observed correspond to the ratios
of the underlying emission factors. Chemical processes are at play in the atmosphere that can lead to a difference between these two variables though. However atmospheric lifetimes of NO$_X$ and CO (respectively 1 to 10 days and 1 to 4 months (Seinfeld and Pandis, 2006)) are too long for these species to be significantly removed during the few hours we focus on. The NO$_X$/CO ratio at emission and concentration ratio are thus expected to be very close. For particulate matter, the discussion is less clear. Secondary PM can form, adding to the emitted PM$_{2.5}$. The nucleation of secondary PM would lead to a NO$_X$/PM$_{2.5}$
concentration ratio smaller than the emission ratio. However, one order of magnitude separates the two regimes we observe, and the time scale we study does not leave much time for secondary PM to become significant, implying a small contribution to the total PM (for instance the apportionment of secondary versus primary aerosols in our baseline simulation varies between 4% and 63% with an average value of 33%).





## 5 Conclusions

In the last decades, decontamination plans in Santiago have mainly focused on decreasing traffic, industrial and residential heating emissions. The chemical footprint of extreme peak events evidenced in this study advocates in favor of also considering more specific and sporadic sources based on cultural habits such as barbecues. Indeed, the mitigation policies currently implemented are helping lowering background pollution levels, but based on our study, cannot prevent extreme $PM_{2.5}$ events from happening while their impact can be significant as well. The "game effect" phenomenon has been hypothesized by lo-

cal authorities but had not been backed up by scientific evidence so far. An analysis of the time characteristics of the events showed that they happen exclusively during soccer games of the national team, played in evenings before a non-working day. The observed concentrations of $NO_X$, CO and $PM_{2.5}$ at that times, compared with the usual levels, allow to trace back the main contribution to fine particles emitted by barbecue cooking. The amount of barbecues cooked during one peak event is estimated and the associated emissions are plugged into a chemistry-transport simulation, leading to the reproduction of the observed

peaks, which is not the case without these emissions. The model then yields appropriate levels, thus comforting the estimated emissions, and allowing to study the evacuation of the $PM_{2.5}$ plume towards the Southwest of the metropolitan area. The more general question of the fate and impacts of particulate matter plumes generated in Santiago is raised.

Although the results provided here seem to have geographically limited implications, the general methodology can be reproduced and benefit to other places in the world. Indeed, using only a limited set of data, with no speciation of particulate matter,

already enables to conclude on the source of extreme events, provided they are tied to cultural habits differing from the usual background air pollution. Such type of sporadic habits are usually ignored by air quality plans implemented by cities, due to the lack of scientific evidence.

In addition, a model-based approach allowed to constrain the estimate of number of barbecues cooked during the final game of the 2016 Copa América (around 100,000). Not only this result can have an informative value for local authorities, but such

a sensitivity analysis can be reproduced and applied to other cases throughout the world in order to estimate the burden on air quality of specific sources.

*Code availability.* The CHIMERE model used can be found at http://www.lmd.polytechnique.fr/chimere/CW-download.php. The WRF model used can be found at http://www2.mmm.ucar.edu/wrf/users/download/get_source.html.

*Data availability.* Surface observation data used in this study are available at https://sinca.mma.gob.cl/index.php/region/index/id/M. HTAP

raw emission inventory can be downloaded at http://edgar.jrc.ec.europa.eu/htap_v2/. Other data can be made available from the corresponding author upon reasonable request.





*Author contributions.* N. Huneeus provided meteorological profiles data and synoptic stations wind speeds used to assess the quality of the simulation. As developers of CHIMERE, L. Menut and S. Mailler supervised the chemistry transport simulations and analyses of the results. R. Lapere performed the data analysis and model simulations, and coordinated the writing of the paper with L. Menut, S. Mailler and N.
Huneeus.

*Competing interests.* The authors declare that they have no conflict of interest.

*Acknowledgements.* N. Huneeus acknowledges the projects FONDECYT Regular 1181139 and FONDAP 15110009.
The authors are grateful to Ricardo C. Muñoz for providing us with the ceilometer backscatter profiles and mixed layer height computations used in the meteorological analysis.





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




| WRF configuration | | CHIMERE configuration | |
|---|---|---|---|
| Coarse domain resolution | 15km | Coarse domain resolution | 15km |
| Nested domain resolution | 3km | Nested domain resolution | 3km |
| Microphysics | WSM3 | Chemistry | MELCHIOR |
| Boundary and surface layer | MYNN | Gas/Aerosol Partition | ISORROPIA |
| Land surface | Noah LSM | Horizontal Advection | Van Leer |
| Cumulus parameterization | Grell G3 | Vertical Advection | Upwind |
| Longwave radiation | CAM | Boundary Conditions | LMDz-INCA + GOCART |
| Shortwave radiation | Dudhia | | |

**Table 1.** WRF and CHIMERE configurations.

| Station: | El Bosque | | | | Independencia | | |
|---|---|---|---|---|---|---|---|
| | MB | NRMSE | R | | MB | NRMSE | R |
| TEMP | 1.3 | 0.34 | 0.84 | : | -1.06 | 0.2 | 0.86 |
| RH | -12.7 | 0.30 | 0.62 | : | -16.6 | 0.31 | 0.7 |
| WS | -0.08 | 0.42 | 0.56 | : | 0.23 | 1.29 | 0.7 |

**Table 2.** Simulation scores for low-level meteorology. June 15[th] to July 15[th] 2016

| Day: | June 27[th] 2016 | | | | June 28[th] 2016 | | | | June 29[th] 2016 | | |
|---|---|---|---|---|---|---|---|---|---|---|---|
| | MB | RMSE | R | | MB | RMSE | R | | MB | RMSE | R |
| TEMP | -0.79 | 2.01 | 0.99 | : | 0.51 | 1.1 | 0.99 | : | 0.83 | 2.4 | 0.95 |
| RH | 7.64 | 14.3 | 0.80 | : | -1.04 | 5.88 | 0.67 | : | -7.47 | 22.2 | 0.82 |
| WS | -1.92 | 3.54 | 0.39 | : | 1.06 | 2.71 | 0.84 | : | -3.63 | 4.38 | 0.99 |
| WD | -2 | 4 | 0.34 | : | -4 | 82 | 0.71 | : | 19 | 85 | 0.48 |

**Table 3.** Simulation scores for meteorological vertical profiles for 3 days at DMC station in Santiago.





| Network: | La Florida | | | | Las Condes | | | | Puente Alto | | |
|---|---|---|---|---|---|---|---|---|---|---|---|
| | MB | NRMSE | R | | MB | NRMSE | R | | MB | NRMSE | R |
| $PM_{2.5}$ | -0.45 | 0.55 | 0.63 | : | 3.31 | 0.62 | 0.62 | : | -3.26 | 0.72 | 0.45 |

**Table 4.** Simulation scores for low-level $PM_{2.5}$ concentrations. June 28[th] to July 15[th] 2016