# Peer review of "Soccer games and record-breaking PM2.5 pollution events in Santiago, Chile."

_Atmospheric Chemistry and Physics, 2019_

## Referee Comment (RC1) · Anonymous Referee #1 · 16 Dec 2019

The manuscript presents the impact soccer games and their related cultural habits may have on air quality in a large city such as Santiago, Chile. Extreme PM2.5 events reaching up to 500 $\mu$g/m3 have been studied, with the traffic and meteorology alone not being able to account for those values, based on the derived chemical signature of NOx/CO and NOx/PM2.5 ratios. When taking into account cooking as a source and given the estimated emission factors from barbeque cooking from different studies, it occurs that the observed ratios during the extreme events of observed PM2.5 concentrations indeed correspond to emissions from barbeques. Tracking back specific events to the dates of observed events it occurs that extreme events are associated with international soccer games involving the Chilean national team, with concentrations being observed in higher intensity during evenings before a non-working day. When the as-

sociated emissions are coupled with a chemistry-transport simulation, observed peaks are highly reproduced, which is not the case without considering the specific emissions. Having reproduced the specific levels, the model then offers the possibility of studying the dispersion of the PM2.5 plume and pinpoint the areas which could be affected by such extreme events.

The paper is well written and easy to follow, and an important point made is that such analysis can be applied in other cases around the globe in order to estimate the burden on air quality of specific sources.

Nevertheless, there are some issues and more thorough discussion should be made in specific sections. Other than that the paper can be recommended for publication after addressing the issues listed below.

General comments:

- There is no mention on what is considered as "background concentrations" neither for PM2.5 nor for the species used for the chemical signature (NOx, CO). Is it below some threshold value?

- There is a lot of mentioning throughout the text about mitigation, decontamination measures etc. and how Chilean authorities should also take into consideration the specific source from this cultural habit, but what possibly can be done in this case? Don't allow barbeques during international soccer games? I agree that cooking may be a very important source during such events, and possibly the contribution of cooking for those 6 hours may well be even more that traffic, thus I was wondering what could be an effective air pollution mitigation policy (Abstract L13)?

- Is there any report/association of the specific events epidemiologically? In the introduction it is mentioned that increases in respiratory emergency and pneumonia visits may occur within 2 days after such a peak, are there any records for the events reported in this paper?

Specific comments:

- P9L145 What are the respective ratios for residential heating by domestic combustion? Are these ratios also quite off as the traffic ones?

- P16L217 It would be interesting to see whether the peak event of the 18th of June also followed a similar dispersion scheme or not; this would imply that only specific areas around Santiago could be impacted during such events.
* * *

---

## Referee Comment (RC2) · Anonymous Referee #2 · 27 Jan 2020

The study investigates the impact national soccer games on air quality in Santiago, Chile. Extreme PM2.5 events have been studied, where drivers like traffic emissions or meteorology cannot explain these high PM values. The study therefore uses observed polltant ratios and show that observed PM2.5 concentrations actually correspond to emissions from barbeques during national soccer games. Emission estimates based on these findings are implemented in a chemistry-transport simulation, which led to nicely reproduction of the observed peaks.

The paper is well written and easy to follow and I would therefore recommend publication of the manuscripts in ACP, given the below minor details are addressed.

Line 2: . . . up to ten-time ABOVE average levels.

[Figure]

Line52: Recommend to write the spatial resolution explicitly in text.

Line 56: What is the spatial and temporal resolution of the FNL data?

Lines 59-60: check the spin-up and the simulation periods, looks like there is a gap of 15 days between the spin up and the actual simulation?

Line 68: Please write the types of the stations: urban, street, etc.?

Line 71: In general please add also relative biases/changes along with the absolute values throughout the text.

Line 83: Mazzeo et al. (2018) used. . ...

Figure 3a: Considering using shading for the period instead of dashed lines for better readability.

Figure 8 is not referred anywhere in the text. Please explain what the shaded areas in Figure 8 represents.

Line 187: . . .(see Section 2 for details). . ..

Line 200: .. would be a total OF 2 tons. . .

---

## Author Comment (AC1) · 26 Feb 2020

**Soccer games and record-breaking PM$_{2.5}$ pollution events in Santiago, Chile**

Rémy Lapere et al.
https://doi.org/10.5194/acp-2019-820

Dear Editor and reviewers,

We acknowledge the reviewers for the time spent to evaluate our work and for their minor revisions. We also acknowledge the Editor and we made all proposed changes in the revised manuscript. Please note that answers are in blue and after each reviewer's remark.

All reviewers' comments were taken into account and are detailed in this letter. Summarizing our answers:

1. Text and Figures were checked and corrected as requested so as to enhance readability.

2. More detailed information on the model configuration, nature of the data used, and usual pollution situations in Santiago were added as per the reviewers suggestions.

3. A study of the dispersion scheme of the second peak event was included to assess whether evacuation patterns differ from one peak to the other.

4. Although comments related to political action and epidemiology are of great importance, and even more justified by the results of the present study, we believe they fall out of the scope of this paper but call for further research in other fields.

Best regards,
Rémy Lapere
February 11, 2020

**1    Reviewer #1**

The manuscript presents the impact soccer games and their related cultural habits may have on air quality in a large city such as Santiago, Chile. Extreme $PM_{2.5}$ events reaching up to $500\mu g/m^3$ have been studied, with the traffic and meteorology alone not being able to account for those values, based on the derived chemical signature of $NO_X/CO$ and $NO_X/PM_{2.5}$ ratios. When taking into account cooking as a source and given the estimated emission factors from barbeque cooking from different studies, it occurs that the observed ratios during the extreme events of observed $PM_{2.5}$ concentrations indeed correspond to emissions from barbeques. Tracking back specific events to the dates of observed events it occurs that extreme events are associated with international soccer games involving the Chilean national team, with concentrations being observed in higher intensity during evenings before a non-working day. When the associated emissions are coupled with a chemistry-transport simulation, observed peaks are highly reproduced, which is not the case without considering the specific emissions. Having reproduced the specific levels, the model then offers the possibility of studying the dispersion of the $PM_{2.5}$ plume and pinpoint the areas which could be affected by such extreme events.

The paper is well written and easy to follow, and an important point made is that such analysis can be applied in other cases around the globe in order to estimate the burden on air quality of specific sources.

Nevertheless, there are some issues and more thorough discussion should be made in specific sections. Other than that the paper can be recommended for publication after addressing the issues listed below.

**General comments:**

There is no mention on what is considered as "background concentrations" neither for $PM_{2.5}$ nor for the species used for the chemical signature ($NO_X$, CO). Is it below some threshold value?

*Answer:*

The use of the word "background" in the manuscript may have been misleading. We actually refer to conditions of pollution other than peak situations, i.e. average conditions for wintertime. For instance Figure 6 in the manuscript is what we would have refered to as background ratios, but is actually more of a regular ratio for the station and season considered. A rewording was made in the revised version of the manuscript so as to get rid of the notion of background and avoid misunderstanding.

There is a lot of mentioning throughout the text about mitigation, decontamination measures etc. and how Chilean authorities should also take into consideration the specific source from this cultural habit, but what possibly can be done in this case? Don't allow barbeques during international soccer games? I agree that cooking may be a very important source during such events, and possibly the contribution of cooking for those 6 hours may well be even more that traffic, thus I was wondering what could be an effective air pollution mitigation policy (Abstract L13)?

*Answer:*

A possibility would be indeed to forbid barbeque cooking when a peak is expected to occur (typically important games during a week-end evening) but this is politically hard to defend. Incentivize the use of electric cooking devices instead of coal could also decrease the emissions as well, much like what is currently done to replace residential wood burning with pellets which emit less pollutants. However, despite the very high interest of the question, such a discussion falls out of the scope of our study, which purpose is not to be prescriptive on a political level. So as to avoid confusion and to make it clearer that policy design is out of scope of this study, a related sentence in the introduction was removed in the new version manuscript.

Is there any report/association of the specific events epidemiologically? In the introduction it is mentioned that increases in respiratory emergency and pneumonia visits may occur within 2 days after such a peak, are there any records for the events reported in this paper?

*Answer:*

Unfortunately, no epidemiological study was found for Santiago assessing the effects on respiratory emergency of the events we are interested in. However, (Ilabaca et al., 2011) estimates the effect of fast increases in $PM_{2.5}$ in Santiago on

respiratory-related emergency visits (REVs), for the years 1995-1996. Although no distinction between PM sources is made, the conclusion of (Ilabaca et al., 2011), as briefly mentioned in the introduction of the manuscript, is that "During the cold months, an increase of $45\mu g/m^3$ in the PM$_{2.5}$ 24-hr average was related to a 2.7% increase in the number of REVs (95% CI, 1.1–4.4%) with a two-day lag". In our case, the increase in concentrations from one day to the other is even higher and sharper, which likely implies an even more dramatic effect on the number of REVs. Given the results presented in our manuscript, such an epidemiological study on the events of 2014 and 2016 would indeed be of important value.

**Specific comments:**

1. P9L145 What are the respective ratios for residential heating by domestic combustion? Are these ratios also quite off as the traffic ones?

   *Answer:*

   Thanks to this relevant remark, the discussion on expected emission ratios for different usual sources of PM was improved, based on emissions ratios provided by HTAP emissions inventory. This allows to be even more confident that we can rule out domestic combustion for residential heating, in addition to the discussion of the expected temporal variations that were discussed in the previous version of the manuscript. This translates into the new version of the manuscript by the following sentences in Section 3.2:

   *Similarly, emission ratios extracted from the HTAP inventory for traffic at the grid point corresponding to Santiago yield 7.5% for NO$_X$/CO and 1750% for NO$_X$/PM$_{2.5}$. This does not match with the PPE signal, especially regarding NO$_X$ and PM$_{2.5}$. For residential heating, the HTAP inventory gives emission ratios at the grid point of Santiago of 19% for NO$_X$/CO and 200% for NO$_X$/PM$_{2.5}$, then again departing from the values observed during PPE. [...] HTAP emission ratios for industry also significantly differ from the PPE situation, with 20% for NO$_X$/CO and 186% for NO$_X$/PM$_{2.5}$.*

2. P16L217 It would be interesting to see whether the peak event of the 18th of June also followed a similar dispersion scheme or not; this would imply that only specific areas around Santiago could be impacted during such events.

   *Answer:*

   The analysis of the dispersion for the second peak event is included in the new version of the manuscript, with the addition of Figure 1 and the adjustment of the associated paragraph which new version can be found hereafter. For the studied year, dispersion indeed follows similar patterns for the two events. However, although the two events studied are similar, that does not imply that the same area will systematically be impacted.

   In the manuscript:
   *Figures 11 and 12 show the difference in PM$_{2.5}$ concentration between the two simulations aforementioned, for the peaks on July 26th and July 18th, respectively. Concentrations at 60m above ground level are considered in order to get rid of the signal of emissions. The simulations result in an evacuation of the particles towards the Southwest of Santiago for both events, a few hours after the onset. Then again the episodes are short-lived in the city, but have impacts on adjacent areas several hours later and several kilometers away from the emission site. For both events the areas impacted by the dispersion of the plume are the South and Southwest regions. Although this result is consistent with mountain-valley circulation, it does not imply that such a dispersion pattern is the only one that can occur during peak events.*

[Figure]

**Figure 1.** *$PM_{2.5}$ concentration difference at 60m above surface between the peak event and baseline simulation in June $18^{th}/19^{th}$. Positive values indicate excess concentrations in the peak event scenario compared to baseline. Map background layer: World Shaded Relief, ESRI 2009.*

**2 Reviewer #2**

The study investigates the impact national soccer games on air quality in Santiago,Chile. Extreme $PM_{2.5}$ events have been studied, where drivers like traffic emissions or meteorology cannot explain these high PM values. The study therefore uses observed polltant ratios and show that observed $PM_{2.5}$ concentrations actually correspond to emissions from barbeques during national soccer games. Emission estimates based on these findings are implemented in a chemistry-transport simulation, which led to nicely reproduction of the observed peaks.

The paper is well written and easy to follow and I would therefore recommend publication of the manuscripts in ACP, given the below minor details are addressed.

**Specific comments:**

1. Line 2: ...up to ten-time ABOVE average levels.

   *Answer:*
   The sentence was corrected accordingly in the new version of the manuscript.

2. Line 52: Recommend to write the spatial resolution explicitly in text.

   *Answer:*
   The following sentence detailing our model resolution was added at the second line of section 2.2:
   *with a coarse domain at 15km spatial resolution comprising most of Chile, and a nested domain focusing on central Chile and centered on Santiago at 3km resolution.*

3. Line 56: What is the spatial and temporal resolution of the FNL data?

   *Answer:*
   The data used for initial and boundary meteorological conditions has a 1° by 1° spatial resolution and 6-hour temporal resolution. This information was added in the new version of the manuscript.

4. Lines 59-60: check the spin-up and the simulation periods, looks like there is a gap of 15 days between the spin up and the actual simulation?

   *Answer:*
   The description of these periods may have been unclear in the previous version of the manuscript. The initial sentence "The simulated period is June 1$^{st}$ to July 15$^{th}$ 2016, with a spin-up period from June 1$^{st}$ to June 15$^{th}$", was replaced by *The simulated period is June 1$^{st}$ to July 15$^{th}$ 2016, with the first 15 days used for spin-up.*

5. Line 68: Please write the types of the stations: urban, street, etc.?

   *Answer:*
   All the monitoring stations of the SINCA network used in this study are representative of urban air quality, according to the associated Chilean regulation that can be found on the SINCA website (https://sinca.mma.gob.cl/index.php/). The urban type of the stations is precised in the section "Observation data" of the new version of the manuscript.

6. Line 71: In general please add also relative biases/changes along with the absolute values throughout the text.

   *Answer:*
   The mean bias on 10m wind speed is additionally described in terms of relative bias in the new version of the manuscript. Observed average levels of $PM_{2.5}$ for the period were also added in order to compare with the bias. This is done in the body of the article directly, tables remaining unchanged.

7. Line 83: Mazzeo et al. (2018) used...

   *Answer:*
   The typing error was corrected.

8. Figure 3a: Considering using shading for the period instead of dashed lines for better readability.

   *Answer:*
   Dashed lines delimiting the period of interest in  Figure 2  were changed to a grey shading, thus improving the readability of the graph.

[Figure]

**Figure 2.** Top: time series of hourly PM$_{2.5}$ concentration between June 1$^{st}$ and August 31$^{st}$ 2016 for the 11 stations of the air quality network of Santiago. Bottom: ratio between hourly PM$_{2.5}$ and average over the summer, zoomed between June 18$^{th}$ and June 28$^{th}$ (shaded period in top graph).

9. Figure 8 is not referred anywhere in the text. Please explain what the shaded areas in Figure 8 represents.

   *Answer:*
   This figure was actually refered to at line 165 of the manuscript to illustrate the correlation between barbeques and soccer games. The caption associated with the figure was improved to enhance understanding as per the reviewer's suggestion. The caption in the new version of the manuscript is now:

   *PM$_{2.5}$ peak events coincidence with soccer games. (a) Hourly PM$_{2.5}$ concentrations at Pudahuel monitoring station (solid blue line), kick-off hours of soccer games (red diamonds) and non-working days (gray shaded areas) - June 2016. (b) same as (a) for June 2015. (c) same as (a) for June 2014.*

10. Line 187: ...(see Section 2 for details)...

    *Answer:*
    The typing error was corrected.

11. Line 200: ...would be a total OF 2 tons...

    *Answer:*
    The sentence was corrected accordingly.

**References**

Ilabaca, M., Olaeta, I., Campos, E., Villaire, J., Tellez-Rojo, M. M., and Romieu, I.: Association between Levels of Fine Particulate and Emergency Visits for Pneumonia and other Respiratory Illnesses among Children in Santiago, Chile, Journal of the Air & Waste Management Association, https://doi.org/10.1080/10473289.1999.10463879, 2011.